

# Anabolic metabolism of autotoxic substance coumarins in plants

Bei Wu, Shangli Shi, Huihui Zhang, Baofu Lu, Pan Nan and Yun A

Key Laboratory of Grassland Ecosystem of Ministry of Education, College of Pratacultural Science, Gansu Agricultural University, Lanzhou, Gansu, China

## ABSTRACT

**Background**. Autotoxicity is an intraspecific manifestation of allelopathy in plant species. The specialized metabolites and their derivatives that cause intraspecific allelopathic inhibition in the plant are known as autotoxic substances. Consequently, autotoxic substances production seriously affects the renewal and stability of ecological communities.

**Methods**. This article systematically summarizes the types of autotoxic substances present in different plants. They mainly include phenolic compounds, terpenoids, and nitrogenous organic compounds. Phenolic coumarins are the main autotoxic substances in many plants. Therefore, we also discuss differences in coumarin types and content among plant varieties, developmental stages, and tissue parts, as well as their mechanisms of autotoxicity. In addition, we review the metabolic pathways involved in coumarin biosynthesis, the key enzymes, genes, and transcription factors, as well as factors affecting coumarin biosynthesis.

**Results**. Coumarin biosynthesis involves three stages: (1) the formation of the coumarin nucleus; (2) acylation, hydroxylation, and cyclization; (3) structural modification. The key enzymes involved in the coumarin nuclear formation stage include PAL, C4H, 4CL, HCT, CAOMT, COSY, F6'H, and CCoAOMT1, and the key genes involved include BGA, CYP450 and MDR, among others. Ortho-hydroxylation is a key step in coumarin biosynthesis and PS, COSY and S8H are the key enzymes involved in this process. Finally, UGTs are responsible for the glycosylation modification of coumarins, and the MaUGT gene may therefore be involved in coumarin biosynthesis.

**Conclusion**. It is important to elucidate the autotoxicity and anabolic mechanisms of coumarins to create new germplasms that produce fewer autotoxic substances.

## INTRODUCTION

Allelopathy refers to the phenomenon *via* which plants directly or indirectly inhibit the seed germination and seedling growth of offspring by releasing specialized metabolites into the environment. Autotoxicity is an intraspecific manifestation of allelopathy in plant species, and is widely distributed in Solanaceae, Cucurbitaceae, Umbelliferae, Araliaceae, Poaceae, and Leguminosae (*Rice, 1971*; *Wang et al., 2022*). Autotoxicity not only obstructs continuous cropping but also affects material circulation, energy flows, system succession, and ecological diversity. Taken together, these impacts can seriously affect the renewal

Corresponding author
Shangli Shi, shishl@gsau.edu.cn

and stability of ecological communities (*Miller, 1983*; *Li et al., 2009a*; *Li et al., 2009b*). Therefore, it is important to study the metabolic mechanisms responsible for the main autotoxic substances produced by plants.

As a form of environmental stress, autotoxicity can be divided into three stages (Fig. S1): (1) biosynthesis of autotoxic substances from donor plants, (2) release of autotoxic substances, and (3) autotoxicity induction in recipient plants. During this process, a donor plant synthesizes different types of specialized metabolites in different concentrations, which are subsequently released into the environment *via* volatilization, leaching, root secretion, or residue decomposition, thereby directly or indirectly inhibiting seed germination and seedling growth of conspecific offspring (*i.e.,* autotoxicity). Thus, the specialized metabolites and their derivatives that cause plant autotoxicity are known as autotoxic substances (*Miller, 1996*; *Rong, 2017*). Studies have reported that the accumulation of autotoxic substances in plants depends on plant growth conditions, likely because autotoxicity is an evolutionary strategy that facilitates plant adaptation to adverse environments. Environmental factors are therefore important determinants of the production and accumulation of autotoxic substances. When plants are exposed to heat, cold, drought, or high salinity, the cellular osmotic balance is disrupted, causing membrane lipid peroxidation, reactive oxygen metabolism disorders, and induction of plant antioxidant defenses and triggering the differential regulation of the synthesis and accumulation of autotoxic substances. Plant genetic background, individual development, and differences in morphology also affect the synthesis and accumulation of autotoxic substances (*Kimber, 1967*; *Om et al., 2002*). At present, the anabolic mechanisms involved in autotoxic substance production are being explored *via* the mining of enzymes, genes, and transcription factors related to the biosynthesis and metabolism of autotoxic substances and by the establishment of genetic transformation systems. Using this information, the problem of autotoxicity can be solved by breeding new cultivars that produce fewer (or no) autotoxic substances (*Chung & Miller, 1995*).

Recent analyses have examined the autotoxic substances produced by plant. These studies have reported that the main specialized metabolites causing plant autotoxicity include coumarins, chlorogenic acid, coumaric acid, hydroxybenzoic acid, caffeic acid, ferulic acid, cinnamic acid, and other phenolic substances (*Zhang & Lin, 2009*; *Tao, 2019*). For some plants, coumarins are the most abundant autotoxic substance, and their impact on autotoxicity is the greatest (*Chung et al., 2000*; *Lu et al., 2007*; *Li et al., 2009a*; *Li et al., 2009b*; *Rong, Shi & Sun, 2016*). Therefore, this article reviews the types of autotoxic substances produced by plants; mechanistic basis of coumarins autotoxicity; metabolic pathways involved in coumarins biosynthesis; key enzymes, genes, and transcription factors involved in coumarins biosynthesis; and factors affecting coumarins biosynthesis. This review can therefore provide a theoretical basis for the further elucidation of the anabolic mechanism of coumarins in plants. In the long run, it is of great significance to create new germplasms with low or no autotoxic substances.

## SURVEY METHODOLOGY

In this article, Sci-Hub (https://sci-hub.wf/), Panda Science (https://panda321.com/), Web of Science (https://www.webofscience.com/wos/alldb/basic-search) and Baidu Academic (https://xueshu.baidu.com/) were used to search for revelant articles. The research keywords included "autotoxicity", "different plants, alfalfa, types of autotoxic substances", "Coumarins classification", "different varieties, parts, growth period, types and content of coumarins", "coumarins autotoxicity", "Factors affecting the synthesis of specialized metabolites", "Abiotic stress, coumarins content", "Environmental stress, different varieties, coumarins accumulation"; "Coumarins synthesis metabolic pathways", "Enzyme genes related to coumarins synthesis". This article aims to clarify the main autotoxic substance of alfalfa, the autotoxicity of coumarins and the research progress of related enzymes and genes involved in its biosynthesis. Therefore, a large number of related studies were collected and screened, and articles with low correlation were excluded.

### Autotoxic substances in plants
#### *Types and contents of autotoxic substances*
In recent years, autotoxins have been isolated and identified from plants in the Liliaceae, Labiatae, Solanaceae, Cucurbitaceae, Gramineae, Umbelliferae, and Leguminosae families (Table S1). These autotoxic substances have been classified into three main groups according to their biosynthetic pathways. They include: (1) phenolic compounds: flavonoids, phenolic acids, coumarins, tannins, and cinnamic acid derivatives, among others. (2) Terpenoids: terpenes, alcohols, aldehydes, ketones, and sesquiterpenes, among others. (3) Nitrogenous organic compounds: alkaloids, glucosinolates, and cyanogenic glycosides, among others. (*Tao, 2019*).

Many studies have identified that phenols as the most autotoxic substances isolated from plants. For example, cucumber (*Cucumis sativus*) root exudates contain benzoic acid, p-hydroxybenzoic acid, vanillic acid, ferulic acid, and other phenolics, and phenolic substances found in wheat (*Triticum aestivum*) include benzoic acid, vanillic acid, coumarin, ferulic acid, and cinnamic acid. Moreover, the content of various phenolic substances is affected by plant variety, location, age, and other factors (*Du, 2006*; *Wang et al., 2018*). In *Angelica sinensis*, analysis of the rhizosphere soil revealed the presence of phenolic compounds such as imperatorin and ferulic acid (*Xin et al., 2019*). Autotoxic substances such as ferulic acid, cinnamic acid, and coumarin are found in the rhizosphere soil of *Paeonia ostii*, and both *Panax notoginseng* and *P. quinquefolius* have been found to contain phenolic autotoxic substances (*Qin et al., 2009*). In addition, it has been reported that the continuous cropping soil of *P. notoginseng* contains p-hydroxybenzoic acid, coumaric acid, syringic acid, and ferulic acid (*Wu, Liu & Zhao, 2014*; *Wu et al., 2014*; *Xiang, 2016*). *Bi, Yang & Gao (2010)* isolated types of phenolic acids found in the fibrous roots of *P. quinquefolius*, and were able to identify syringic acid, vanillin, p-coumaric acid, and ferulic acids. In addition, previous studies have also reported that leguminous plants contain various autotoxic substances. For example, *Asaduzzaman & Asao (2012)* found that fatty acids (such as succinic acid, lactic acid, and malic acid), and phenolic acids (such as benzoic acid, p-hydroxyphenylacetic acid, and vanillic acid) were the main
autotoxic substances in the root exudates of broad bean (*Vicia faba*). Previous studies have shown that the accumulation of phenolic acids such as phenylacetic acid, cinnamic acid, 4-hydroxybenzoic acid, and phthalic acid in soils was an important impediment for the cultivation of continuous cowpea (*Vigna unguiculata*) (*Huang, 2010*). Furthermore, the rhizosphere soil of peanut (*Arachis hypogaea*) crops has been found to contain p-hydroxybenzoic acid, vanillic acid, coumaric acid, coumarin, and other phenolic autotoxins, and alfalfa is able to secrete coumarin, coumaric acid, p-hydroxybenzoic acid, caffeic acid, chlorogenic acid, and ferulic acid (*Huang et al., 2013*; *Rong, 2017*). *Ghimire et al. (2019)* isolated salicylic acid, p-hydroxybenzoic acid, scopoletin, quercetin, and other autotoxic substances from alfalfa leaves.

Other studies have analyzed the content of autotoxic substances and their autotoxicity to identify the most important autotoxic substances present in plants. Many such studies have reported that coumarin was the most abundant of the major phenolic autotoxic substances measured in the aboveground and underground parts of different varieties of alfalfa (*Li et al., 2009a*; *Li et al., 2009b*; *Rong, Shi & Sun, 2016*). *Lu et al. (2007)* used high performance liquid chromatography (HPLC) to determine the content of phenolic autotoxins in alfalfa, and found that coumarin levels were relatively high. It has also been reported that of the phenolic compounds that are autotoxic toward alfalfa, coumarin is the most abundant and exerts the strongest inhibitory effect (*Hegde & Miller, 1992*). *Tao et al. (2019)* found that exogenous coumarin, caffeic acid, and their mixture could inhibit normal root morphology of alfalfa, and the comprehensive inhibitory effect of the three was as follows: coumarin >mixture >caffeic acid. *Zheng, Shi & Ma (2018)* have also reported that four substances (*i.e.,* cinnamic acid, hydroxybenzoic acid, coumarin, and tricin) could inhibit the growth of alfalfa to different degrees, but that the autotoxicity of coumarin was the strongest. *Wu, Liu & Zhao (2014)* showed that coumarin was the most abundant allelochemical found in *Melilotus officinalis*, and that the higher the coumarin content, the stronger the allelopathic inhibitory effect. Therefore, the in what follows we will systematically summarize recent research on the autotoxic substance coumarin.

### Classification of coumarins

Coumarins are a class of specialized metabolites that are derived from phenylpropanoid metabolic pathways (*Luo, 2017*). Coumarins can be divided into five categories according to their different substituent structures (Fig. S2): (1) simple coumarins: coumarins with hydroxyl, isopentenyl, methylenedioxyl, and methoxyl substituents on the benzene ring and the hydroxyl group at position 7 not forming a furan or pyran ring with substituents at position 6 or 8, *i.e.,* scopoletin, umbelliferone, and aesculin. (2) Furanocoumarins: coumarins with a furan ring attached to the coumarin nucleus, *i.e.,* psoralen lactone. (3) Pyranocoumarins: coumarins with a 2,2-dimethylpyran ring structure formed by the cyclization of the isoprenyl group at the C6 or C8 position of the coumarin nucleus with an orthophenolic hydroxyl group, *i.e., Angelica sinensis* and pepper lactone. (4) Isocoumarins: 1H-2-benzopyran-1-one, such as bergenin. (5) Other coumarins: monocoumarins with substituents on the coumarin nucleus, coumarin dimers, trimers, and others, such as alfalfa phenol. Simple coumarins are the most commonly identified type and contain hydroxyl,

isopentenyl, methylenedioxyl, and methoxyl substituents on the benzene ring (*Luo, 2017*; *Duan et al., 2021*).

To date, coumarins have been found in >330 genera from 74 families, including Solanaceae, Umbelliferae, and Rutaceae. In addition, various coumarin compounds have been isolated and identified from different plant stems, leaves, flowers, seeds, roots, and root exudates. For example, umbelliferone, scopoletin, and daphnoretin have been isolated and identified from *Stellera chamaejasme*, and imperatorin, isoimperatorin and oxypeucedanin have been detected in *Angelica dahurica*, *Glehnia littoralis*, and *Psoralea corylifolia* (*Liang, 2005*; *Zhou et al., 2021*; *Zhang, 2012*). Furthermore, a study on *Ipomoea cairica* reported that it contains umbelliferolactone and scopolaminolactone, and generates the highest levels in the summer (*You et al., 2014*). Moreover, the peel of *Citrus maxima* contains 7-methylcoumarin, 7-methoxycoumarin, 7-ethoxy-4-methylcoumarin, vinegar nitrate coumarin, and several others; in this species, 7-methoxycoumarin is the most abundant (*Hao, 2019*). Compounds with a simple structure, such as coumarin, scopoletin, and umbelliferone lactone, are widely distributed in many plants in Compositae, Leguminosae, Umbelliferae, and Gramineae. The types of coumarins found in legumes such as *Melilotus officinalis* include coumarin, umbelliferin, scopoletin, 3-hydroxycoumarin, and dihydrocoumarin (*Tang & Fan, 2012*). Alfalfa contains biscoumarins and coumarin-3-4 furan derivatives, including alfalfa phenol, alfalfa lactone, and estrogenic lactone (Table S2) (*Yin & Qin, 2008*). Thus, simple coumarins are the most abundant class of coumarins isolated and identified in plant species.

### Distribution of coumarins in plants

Recent studies have focused on the differences in coumarins type and content in different plant species. Qualitative and quantitative determination of coumarin compounds have revealed that coumarins type and abundance is affected by cultivar, plant organ or part, developmental stage, and environmental conditions (Table S3). *Nie & Zhao (2021)* determined the coumarins content of different parts of six types of pomelo (*Citrus grandis*) fruits and reported that among all coumarins, isoimperatorin and isomeranzin contents were the highest in the flavedo and albedo, while in the pulp, bergaptol and 6′, 7′-epoxybergamottin were more abundant. Under the same conditions, the coumarin content in the leaves of young *Mikania glomerata* plants was considerably higher than that in adults, and the coumarin content of leaves and stems was also higher (*Castroem et al., 2007*). Another study detected 34 coumarins, mainly pyranocoumarins, in *Peucedanum praeruptorum*, and reported the highest coumarin content to be present in the secretory canal and the lowest coumarin content to be in the secondary xylem. Moreover, the levels of seven coumarins decreased after bolting, with the most substantial decrease recorded for pyranocoumarin (*Chen et al., 2019*). Another study reported that the root extract of *Chrysanthemum segetum* is rich in 7-methoxycoumarin, and angelica lactone was the major constituent extracted from the aerial parts of the plant (*Ochocka et al., 1995*). *Gao & Li (2023)* detected 37 and 36 coumarins from the primary and lateral roots of *A. dahurica*, respectively. Coumarins found in the primary roots were mainly concentrated in the periderm, cortex, and phloem, whereas the coumarins in lateral roots were mainly

concentrated in the phloem. The furanocoumarins of *Levisticum officinale* were mainly distributed in the roots (*Olennikov, 2023*). Furthermore, the relative abundance of different autotoxic compounds in different parts of the alfalfa plant reportedly reflect the following pattern: leaf >seed >root >flower >stem (*Chon et al., 2002*; *Wang, 2008*; *Wang, Wu & Zhao, 2017*). In addition, the autotoxicity of alfalfa at different growth stages was as follows: podding stage >early flowering stage >bud stage >branching stage >seedling stage (*Li et al., 2009a*; *Li et al., 2009b*). The same study also reported differences in coumarin content between different stubbles in the same year, the pattern of which was as follows: third stubble >second stubble >first stubble (*Yuan, 2008*).

## Autotoxicity of coumarins

Coumarins not only play an important role in plant growth and development, where they act as plant signal molecules and phytoalexins, but also exhibit medicinally useful functions, such as anticancer, antivirus, hypoglycemic, antihypertensive, and neuroprotective properties. As a signal molecule for information transfer in plants, it can delay seed germination by inhibiting abscisic acid catabolism (*Luo, 2017*). One study reported that when *Arabidopsis thaliana* grows in an Fe-deficient environment, coumarin synthesis considerably increases. Moreover, when afflicted by diseases and insect pests, plants can synthesize and accumulate substantial amounts of various coumarins (*Robe et al., 2021*; *Dutsadee & Nunta, 2008*; *Sun et al., 2014*). For example, scopoletin and its glucoside are involved in plant resistance to salicylic acid stress (*Pastırová, Repčák & Eliašová, 2004*). However, the production and release of coumarins not only exerts an allelopathic effect on surrounding plants but also inhibits the seed germination and seedling growth of their own offspring, which exhibits strong autotoxicity (*Razavi, 2011*). In addition, studies on the autotoxicity mechanisms of coumarin have shown that autotoxicity not only affects the seed germination and seedling growth of alfalfa but can also affect the expression of genes and activity of enzymes related to photosynthesis, osmotic regulation, antioxidant functions, and hormonal regulation (Fig. S3) (*Miller, 1996*).

### Effects of coumarins on seed germination and seedling growth

The seed germination stage is a primary stage of plant growth and is highly sensitive to changes in the external environment. This process therefore is the first step during which plant autotoxicity can be observed. Autotoxicity can significantly inhibit seed germination of the next generation, affect the formation and function of seedling roots, thereby reducing the effective absorption and utilization of nutrients (*Yang et al., 2021a*; *Yang et al., 2021b*). Coumarin has been found to be a strong inhibitor of seed germination in durum wheat (*Triticum durum*), and after coumarin treatment of wheat seeds, one study reported that the seed germination rate, electrolyte retention and oxygen consumption of affected seeds decreased significantly (*Abenavoli et al., 2001*; *Chuah, Tan & Ismail, 2013*). Another study of Italian ryegrass (*Lolium multiflorum*) by *Yao et al. (2017)* found that coumarin can destroy the structure of the cell membrane in cells within the seed endosperm. It can thereby significantly inhibit seed germination and early seedling growth. In addition, coumarin has been found to significantly inhibit alfalfa seed germination rate, germination potential,

radicle length, and germ length (*Aliotta et al., 1993*; *El-Shahawy & Abdelhamid, 2013*; *Li et al., 2022*). Moreover, higher concentrations of coumarin were negatively correlated with alfalfa root length and branch number (*Hegde & Miller, 1992*).

### Effects of coumarins on plant physiology

*Photosynthesis inhibition.* The damage caused by autotoxins to photosynthetic capacity is mainly manifested as damage to the integrity of cell membranes, which limits the intensity of plant stomatal respiration and promotes the decomposition of chlorophyll. Taken together, these effects can significantly reduce the photosynthetic rate and chlorophyll content of seedlings (*Wu et al., 2004*; *Rokem, Lantz & Nielsen, 2007*). It has been reported that a coumarin solution can reduce the chlorophyll content of *Sorghum sudanense* seedlings, change the chloroplast to a round ball shape, and cause mitochondrial deformation. At a concentration of 500 mg $\times$ kg$^{-1}$, the cell structure was completely destroyed (*Wang, Wu & Zhao, 2017*). In contrast, *Ahrabi, Enteshari & Moradshahi (2011)* found that low concentrations of coumarin treatment had little effect on the chlorophyll content of canola (*Brassica campestris*), while high concentrations significantly reduced seedling biomass and chlorophyll content. In alfalfa, coumarin has been found to reduce photosynthetic efficiency by inhibiting stomatal conductance and intercellular $CO_2$ concentration. At 96 h after coumarin exposure, Chla, Chl(a+b), Chl(a/b), and carotenoid content were all significantly reduced, and photosynthesis was inhibited (*Li et al., 2022*). Coumarin treatment was also found to significantly reduce the biomass, photosynthetic pigmentation, and soluble protein content of alfalfa twigs and roots (*Chen, Zhang & Wang, 2011*).

*Inhibition of osmotic regulators.* Low temperatures, drought, salinity, and autotoxicity affect normal plant growth and development. Moreover, under certain environmental stress conditions, plant cells actively accumulate solutes, reduce osmotic and water potential, maintain turgor pressure, change the osmotic regulator content, and thereby resist external stress *via* osmotic regulation (*Hetherington & Woodward, 2003*; *Fang & Xiong, 2015*). Soluble sugars and proteins and proline are important osmotic regulators in plants. Soluble sugars and proteins maintain low osmotic potential levels within plant cells and protect the cell structure against damage caused by environmental stresses (*Zhang et al., 2007*; *Wang, Wu & Zhao, 2017*). Proline is the most water-soluble amino acid, and the accumulation of free proline can resist environmental stress (*Szabados & Savoure (2010)*).

Autotoxicity can affect plant protein synthesis and metabolism, and inhibit amino acid transport by reducing the integration of DNA and RNA, significantly decreasing plant soluble protein content (*Li et al., 2014*; *Huang et al., 2012*). Reportedly, the treatment of annual ryegrass (*L. multiflorum*) seedlings with different concentrations of an exogenous coumarin aqueous solution increased starch content and the emergence of starch granule complexes. With increasing coumarin concentration, the number of compound amyloplasts increased and soluble sugar content decreased (*Wang, Wu & Zhao, 2017*; *Tao, 2019*). In another study, *Li et al. (2022)* treated alfalfa seedlings with coumarin, reporting that coumarin treatment for 24 h considerably reduced soluble protein content in the roots of alfalfa seedlings and that the soluble protein content substantially increased by the 72-

and 96-h post-treatment time points. Finally, the authors also revealed that soluble sugar content first increased and then decreased with prolonged coumarin treatment times.

*Reduced plant antioxidant capacity and increased membrane lipid peroxidation.* During normal plant growth and development, the production and scavenging of free radicals are in dynamic equilibrium. However, the production of autotoxins reduces the activity of antioxidant enzymes such as peroxidase (POD), superoxide dismutase (SOD), and catalase (CAT), and causes the accumulation of reactive oxygen species such as $O^{2-}$, $H_2O_2$, and $OH^-$. The result is the peroxidation or degradation of unsaturated fatty acids in the cell membrane, which gradually decompose into small molecules such as malonaldehyde (MDA). This in turn changes the permeability of the cell membrane (*Luo, 2006*). Phenolic autotoxins damage plants mainly by disrupting the redox balance (*Hong et al., 2008*). *Song et al. (2006)* studied the effects of different concentrations of coumarin on the activity of antioxidant protective enzymes in alfalfa seedlings and reported that substantially coumarin inhibited the activity of SOD, POD, and CAT in seedlings, causing considerably increased in the MDA content. In addition, coumarin treatment also caused ROS accumulation in the root cells of alfalfa seedlings, inhibited the activity of most antioxidant enzymes, and decreased the levels of antioxidants and osmotic regulators, thereby decreasing the ability of alfalfa roots to scavenge ROS and leading to an intracellular redox imbalance and ultimately to oxidative damage in alfalfa seedling roots. Specifically, this study concluded that coumarin caused oxidative damage to alfalfa seedlings by increasing $H_2O_2$, $O^{2-}$, and MDA content and decreasing the activities of SOD and glutathione reductase (*Li et al., 2022*).

*Coumarins affect plant hormone levels.* Endogenous hormones regulate plant growth and morphogenesis. Stress factors such as autotoxicity can affect normal plant physiological processes by reducing the physiological activity of hormones or by causing them to lose their activity (*Wang et al., 2010a*; *Wang et al., 2010b*). For example, phenolic compounds reportedly affect the decomposition of gibberellin and indoleacetic acid (*He & Lin, 2001*). In addition, autotoxic substances can also affect plant growth and development of plants by changing or disruption hormonal balance. For example, increased *Lepidium draba* extract concentration resulted in decreased concentrations of indoleacetic acid, zeatin and gibberellin in *Zea mays* seedlings and caused higher accumulation of abscisic acid (*Kaya et al., 2015*). Similarly, a water extract of sunflower (*Helianthus annuus*) leaves increased abscisic acid content in white mustard (*Sinapis alba*) seeds, thereby reducing ethylene content by affecting the activity of ACC synthase and ACC oxidase. This in turn inhibited seed germination and seedling growth (*Gniazdowska-Piekarska, Oracz & Bogatek-Leszczyńska, 2007*). In another study, *Wang (2018)* investigated the mechanism of coumarin on annual ryegrass and found that coumarin inhibits the secretion of gibberellin and auxin but promotes the secretion of abscisic acid, thereby altering the endogenous hormone regulation of the plant. Finally, in another study, coumarin treatment of alfalfa seedlings stimulated the expression of the key genes NCED, ZEP, and BG, all of which

are involved in the abscisic acid synthesis pathway. The same authors confirmed that this treatment increased abscisic acid content in alfalfa roots (*Tao, 2019*).

## Anabolic metabolism of coumarins
### Coumarin biosynthesis pathways

Coumarins are a class of specialized metabolites derived from phenylpropanoid metabolic pathways. The primary metabolic process involved in coumarin biosynthesis is based on photosynthetic products, which are subsequently converted into glucose by the sucrose–starch metabolism. Glucose produces phosphoenolpyruvic and erythrose-4-phosphate *via* the glycolysis and pentose phosphate pathways (*Li, 2010*). These two substances then work together to enter the shikimic acid pathway, which is catalyzed by 3-deoxy-D-arabino-heptulosonate-7-phosphate synthase (DAHPS), 3-dehydroquinate synthase (DHQS), 3-de-hydroquinic acid dehydrase (DHD), shikimate dehydrogenase (SDH), and shikimate kinase (SK) to produce shikimic acid. Shikimic acid is subsequently used as the substrate to form phenylalanine *via* the transamination of chorismate and prephenic acid, thus entering the secondary metabolic process (*i.e.,* phenylpropanoid metabolism) (*Fu et al., 2021*). The formation of photosynthetic primary products into coumarins is also affected by physiological and biochemical processes such as carbon and nitrogen metabolism, which provide carbon and amino groups, respectively, for coumarins synthesis (*Ruan, Haerdter & Gerendás, 2010*; *Liu, 2016*), They are therefore key factors regulating the accumulation of coumarins (*Wang, 2021a*; *Wang, 2021b*; *Wang, 2021c*). Through transcriptomic analysis of sweetclover germplasms containing different coumarin contents, *Luo et al. (2017)* reported that most differentially expressed genes were enriched in carbon metabolism related pathways. These included the photosynthetic carbon assimilation, glycolysis, and starch-sucrose metabolic pathways, most of which were found to be strongly upregulated in germplasms with high coumarin content. Furthermore, the photosynthetic capacity of *Heracleum moellendorffii* was reportedly the strongest in its middle growth stage, during which its primary metabolism was active, carbohydrate synthesis was active, and coumarin accumulation was promoted. Moreover, changes in coumarin content were positively correlated with the levels of soluble sugar and starch. Finally, the relationships between protein levels and coumarin synthesis is complex: a competitive relationship or an abundance of protein in the plant body can both cause coumarin content increase owing to competition for the same substrate to promote the conversion of photosynthetic products (*Sun, 2019*).

The phenylpropanoid metabolic pathway consists of both a common pathway as well as several branch pathways (*Ou Yang & Xue, 1988*). The common pathway starts with phenylalanine formed by the shikimic acid pathway, followed by phenylalanine ammonia lyase (PAL), cinnamic acid 4-hydroxylase (C4H), and 4-coumaric acid-CoA ligase (4CL), which form important intermediates involved in phenylpropanoid metabolism, *i.e.,* cinnamic acid, p-coumaric acid, and p-coumaroyl-CoA (*Du et al., 2005*; *Fraser & Chapple, 2011*). This step represents the core reaction of secondary metabolite biosynthesis and provides precursors for the biosynthesis of all downstream metabolites (*Xia et al., 2017*; *Dong & Lin, 2021*). Downstream pathways include multiple branches involved in

phenylpropanoid metabolism, including the coumarins synthesis pathways, lignin synthesis pathway, flavonoid synthesis pathway, and proanthocyanidin-specific synthesis pathway (*Hou, 2021*; *Ge, Xin & Tian, 2023*).

The coumarin synthesis pathway is shown in Fig. S4. Here, phenylalanine is converted to cinnamic acid under the action of PAL. The corresponding synthesis pathway differs depending on the specific type of coumarin being synthesized. Unlike the majority of coumarins, coumarin itself is formed by the successive action of cinnamic acid-2-hydroxylase (C2H) and $\beta$-glucosidase (BGLU) (*Stoker & Bellis, 1962*). Another branch pathway features cinnamic acid forming coumaroyl-CoA under the action of C4H and 4CL. This then generates different types of coumarins *via* catalysis facilitated by specific enzymes. P-coumaric acid, isoferulic acid, ferulic acid, and caffeic acid are the intermediates in this complex pathway (*Kindl, 1971*).

### Enzymes and genes related to coumarins synthesis

There are three main steps in the formation of coumarins: (1) coumarin nucleus formation, (2) acylation, hydroxylation, and cyclization, and (3) structural modification. To date, numerous enzymes and genes involved in coumarins synthesis and metabolism have been identified; these include lyases, transferases, ligases, oxygenases, and reductases. Moreover, many of these enzymes are members of gene superfamilies, including the cytochrome P450 monooxygenase, 2-ketoglutarate-dependent dioxygenase, NADPH-dependent reductase, and type III polyketide synthase (PKSIII) gene families (*Turnbull et al., 2004*; *Lu et al., 2006*; *Ververidis et al., 2007*; *Ferrer et al., 2008*; *Wang et al., 2013*). In addition to the genes encoding specific enzymes, several regulatory factors have been identified. For example, MYB is considered to be a major regulatory factor and plays a key role in regulating genes involved in the phenylpropanoid biosynthesis pathway (*Xu, Dubos & Lepiniec, 2015*).

*Enzymes and genes related to coumarin nucleus formation.* Most research into the enzymes involved in coumarins biosynthesis has focused on the first stage, *i.e.*, coumarin nucleus formation. Many studies have confirmed that PAL, C4H and 4CL are the three most critical enzymes active at this stage. PAL catalyzes the hydrolysis of phenylalanine to produce cinnamic acid. Moreover, as PAL activity increases, it promotes the accumulation of coumarin (*Liu et al., 2018*). It has been found that overexpression of AgC4H or AgPAL promotes the production of decursinol angelate in *Angelica gigas* (*Park, Park & Park, 2012*). C4H is a cytochrome P450 monooxygenase complex that catalyzes the hydroxylation of the 4-position of cinnamic acid to produce coumaric acid. Moreover, C4H is highly active in several plant tissues (*Potter et al., 1995*; *Nedelkina et al., 1999*; *Wang et al., 2010a*; *Wang et al., 2010b*). In *P. sativum*, *Apium graveolens*, and *A. thaliana*, C4H is a protein encoded by a single gene, whereas in *Zea mays* it is a protein encoded by a small multigene family (*Koopmann, Logemann & Hahlbrock, 1999*; *Betz, McCollum & Mayer, 2001*). 4CL uses cinnamic acid and coumaric acid as catalytic substrates to promote the production of specialized metabolites including coumarins, flavonoids, and lignin. *Liu et al. (2018)* found that the content of coumarin in *Artemisia argyi* roots was higher than in other tissues; this was due to high levels of 4CL driving the catalytic conversion of p-coumaric acid.

In addition to the study of PAL, C4H, and 4CL, recent studies have also examined other key enzymes and genes involved in coumarins synthesis. For example, key enzymes related to coumarins biosynthesis have been revealed by transcriptomic sequencing of different tissues of *A. dahurica*. These key enzymes include hydroxycinnamoyl CoA shikimate (HCT), p-coumarate 3-hydroxylase (C3H), and caffeic acid catechol-o-methyltransferase (COMT), which interact with each other to regulate coumarins synthesis. Here, COMT catalyzes caffeic acid to obtain hydroxymethyl and ferulic acid (*Xu, Dubos & Lepiniec, 2015*). In one study of *A. dahurica*, a total of 283 unigenes were annotated to the MYB family, of which four MYB TFs were enriched in the phenylpropanoid biosynthesis pathway and therefore may be involved in coumarins biosynthesis. Moreover, MYB72 may regulate the synthesis of scopoletin in roots (*Stringlis, De Jonge & Pieterse, 2019*). *Shi et al. (2020)* also analyzed transcriptomic data from different tissues of *Cnidium monnieri* and identified a large number of genes encoding enzymes related to coumarin biosynthesis. In that study, unigenes encoding HCT were highly differentially expressed in flowers, while unigenes encoding BGLU and F6'H1 were significantly more expressed in roots than in other tissues. BGLU and F6'H1 are both key enzymes in coumarins biosynthesis, since both enzymes play an important role in regulating the metabolism of coumarins, and their gene expression levels can therefore affect coumarins accumulation (*Poulton, McRee & Conn, 1980*; *Yang et al., 2015*). *Yang et al. (2021a)*; *Yang et al. (2021b)* used a metabolomics analysis to find that *Bupleurum chinense* contains five coumarins, including, esculetin, aesculetin, psoralen, and angelica. A correlational analysis of coumarin content and transcriptional changes identified 39 differentially expressed genes from five gene families involved in coumarin synthesis, belonging to the HCT, COMT, 4CL, F6'H and psoralen synthase (PS) gene families, respectively. Further correlation network analysis showed that PS genes (*i.e.,* Bc09210 and Bc12711), COMT genes (*i.e.,* Bc14830, Bc17539, Bc28792, and Bc32856) and HCT genes (*i.e.,* Bc32814 and Bc10245) may be important factors affecting coumarin synthesis. In addition, a combined transcriptomic and metabolomic analysis of *P. praeruptorum* identified predicted CYP450 and MDR genes that may also be involved in the biosynthesis and transport of coumarins (*Zhao et al., 2015*).

### Enzymes and genes related to acylation, hydroxylation, and cyclization

Previous studies have shown that the core structure of coumarins is formed by the orthohydroxylation of cinnamic acid, the trans/cis isomerization of side chains, and by lactonization. Orthohydroxylation is a key step in coumarins biosynthesis. Research on o-hydroxylase involved in coumarins biosynthesis has mainly focused on P450-dependent reactions (*Gestetner & Conn, 1974*). After isopentenylation at the 6 or 8 position of umbelliformis lactones, the hydroxylation process involves several cytochrome P450 monooxygenases, one of which PS has been successfully cloned and identified as the key P450 for psoralen formation (*Hamerski & Matern, 1988*; *Stanjek & Boland, 1998*; *Kai et al., 2008*). In the 1980s, the existence of PS was first demonstrated by cell cultures of *Ammi majus* and by precursor feeding experiments. However, it was not until 2007 that PS was identified in *A. majus* Subsequently, CYP71AJ3 and CYP71AJ2 were found in *Angelica valida* and *Apium graveolens*, respectively (*Larbat et al., 2007*). It has been

found that furanocoumarins present in *P. praeruptorum* were synthesized by enzyme-catalyzed reactions involving umbelliferone. Umbelliferone is known to be first catalyzed by 6-isopentenyl transferase or 8-isopentenyl transferase to generate isoviologen (*Karamat et al., 2014*; *Munakata et al., 2016*). In addition, another study used HPLC-ESI-MS results to demonstrate that the PS CYP71AJ49 can catalyze the conversion of isoviologen to psoralen, and that the angelicin synthase CYP71AJ51 can catalyze the conversion of precursors to angelicin (*Jian et al., 2020*). Furthermore, *Vanholme et al. (2019)* proved that coumarin synthase (COSY) is another key enzyme involved in coumarins biosynthesis. Recombinant COSY can produce umbelliferone, kaempferol, and scopoletin from their respective O-hydroxycinnamoyl-CoA thioesters *via* trans-cis isomerization and lactonization. *Karamat et al. (2014)* had previously identified a membrane-bound prenyltransferase known as PcPT in parsley (*Petroselinum crispum*). PcPT has strict substrate specificity toward umbelliferone and dimethylallyl diphosphate, and shows a strong preference for the C6 position of the prenylated product (demethyllipoprotein). This resulted in the production of furancoumarin; thus PcPT promoted the synthesis of furanocoumarin in parsley. Moreover, the introduction of PcPT into the coumarin-producing plant *Ruta graveolens* resulted in increased consumption of endogenous umbelliferone. In addition, an enzyme specific for bergamot phenol o-methylation (BMT) was found in methoxylated coumarin. Various types of methoxylated coumarins suggested that at least one unknown enzyme was involved in the o-methylation of other hydroxylated coumarins. *Zhao et al. (2019)* conducted transcriptomic and metabolomic analyses of *P. praeruptorum*, and found that an enzyme similar to caffeic acid o-methyltransferase (COMT-S) was involved in catalyzing the hydroxylated coumarins in *P. praeruptorum*. They then identified the o-methylation steps involved in coumarins biosynthesis. It is also known that *A. thaliana* can accumulate scopoletin and its $\beta$-D-glucopyranoside (scopolin) in its roots. *Kai et al. (2006)* used *A. thaliana* to study scopolin biosynthesis. They showed that caffeoyl-CoA-O-methyltransferase (CCoAOMT1) can convert caffeoyl-CoA into feruloyl-CoA, a key precursor of scopolamine biosynthesis. Before the formation of the scopoletin ring, F6'H1 is a key enzyme involved in the orthohydroxylation of feruloyl-CoA, a precursor involved in the synthesis of coumarin, scopoletin and scopoletin derivatives in *A. thaliana*. It belongs to the Fe (II)-and 2-ketoglutarate-dependent dioxygenase (2OGD) family. Moreover, the biosynthesis of scopoletin has been shown to be strongly dependent on CYP98A3 due to the 3'-hydroxylation of p-coumaric acid in the phenylpropanoid pathway, which is catalyzed by cytochrome P450 (*Schoch et al., 2001*). In addition, *Siwinska et al. (2018)* purified the heterologous expression of the At3g12900 protein in *Escherichia coli*, thereby demonstrating that it participated in coumarin biosynthesis *via* the hydroxylation of the scopoletin-8-hydroxylase (S8H) C8 site; they also noted that this protein converted scopoletin into fraxetin.

*Enzymes and genes involved in structural modification.* Many studies have shown that glycosylation of plant specialized metabolites is catalyzed by family 1 UDP-dependent glycosyltransferases (UGTs) (*Bowles et al., 2005*), UGTs convert aglycones into more stable, bioactive, and structurally diverse glycosylated derivatives, such as coumarins,

flavonoid, terpenoids and steroids (*Wilson, Wu & Tian, 2019*). Coumarins are modified to glycosylated forms by the activity of UGTs, such as scopolamine and aesculin (*Chong et al., 2002*). Glycosylated coumarins are then stored in vacuoles. In response to various stress reactions, the destruction of cells puts glycosylated coumarins in contact with $\beta$-glucosidase (BGLU) in the cytoplasm, which catalyzes the hydrolysis of the glycosidic bond between carbohydrates and the coumarin core structure to produce biologically active coumarin aglycones, such as scopoletin and kallidinogen (*Morant et al., 2008*; *Ahn et al., 2010*). A total of 189 MaUGT genes have been identified in the genome of *Melilotus* spp. Of these, 16 MaUGT genes have been found to be differentially expressed in low- and high-coumarin genotypes of *Melilotus*, and may therefore be involved in coumarin biosynthesis (*Duan et al., 2021*). *Wu et al. (2022)* identified candidate genes related to coumarin biosynthesis by selective scanning analysis. One BGLU gene cluster involved in coumarin biosynthesis was identified by combining genomic, BSA, transcriptomic, metabolomic, and biochemical analyses. The function of MaBGLU1 was verified by overexpression in *Melilotus*, heterologous expression in *E. coli* and yeast two-hybrid experiments. *Xu et al. (2021)* cloned a new UGT from Cistanche tubulosa, and extensive *in vitro* enzyme assays found that CtUGTi could catalyze the glucosylation of coumarins umbelliferone 1, esculetine 2, and hymecromone 3.

### Factors affecting coumarins synthesis

The environments in which plants live are not always ideal, and plants are therefore often exposed to forms of environmental stress during growth and development. To adapt to environmental changes, plants produce specialized metabolites in response to environmental stimuli that affect the expression of biosynthetic genes. Specialized metabolites involved in plant defense systems as allelochemicals improve their competitiveness, but can also exert autotoxic effects on seed germination and seedling growth (*Costa et al., 2012*). Although specialized metabolites have no direct effect on plant survival, they can inhibit the biosynthesis of developmental substances in offspring, prevent the synthesis of specialized metabolites in specific environments, or significantly increase metabolite content (*Akula & Ravishankar, 2011*). Studies have shown that the concentration of autotoxic substances depends both on growth conditions and metabolic pathways. Moreover, their transport and storage at the final location are also affected by physiological and cellular factors. Finally, plant development factors also affect the initiation and differentiation of specific cell structures (*Broun et al., 2006*), and the concentrations of different factors are influenced on the genetic level. Genes involved in biosynthetic pathways are generally controlled on the transcriptional level by a variety of transcription factors, which play an important role in regulating the concentration, accumulation, and biosynthesis of various autotoxic substances (*Naghiloo et al., 2012*). Therefore, the factors affecting the content of coumarins in plants include environmental factors as well as individual plant factors (*i.e.,* genetic factors, individual developmental factors, and tissue- or organ-specific factors) (Fig. S5). These four main factors play important roles in influencing the accumulation of autotoxic substances and we discuss each below.

*Environmental factors.* Environmental conditions are decisive factors for the synthesis and accumulation of autotoxic substances. Some environmental factors are biological factors. For example, plants activate various metabolic pathways to resist the attack of pathogens such as fungi, viruses, bacteria, and nematodes, resulting in the synthesis and increase in concentration of various autotoxins (*Wojakowska et al., 2013*). Plants are also affected by abiotic environmental factors; for example, they interact with the surrounding environment during growth and development and are therefore exposed to differences in water, light, temperature, soil, and chemicals (*i.e.,* minerals or fertilizers). When environmental conditions permit, plants can grow, develop, and survive. However, when the abiotic components are excessive or insufficient, such as under high temperature, cold, drought, metal ion, or nutrient deficiency stress conditions, plants can produce specific autotoxic substances (*Gouvea et al., 2012*). Moreover, when these substances accumulate at high concentrations, they can seriously inhibit seed germination and seedling growth of the progeny (*Radušiene, Karpavičiene & Stanius, 2012*). In plants, autotoxic substances also exhibit great structural diversity. They can be produced in response to different biotic or abiotic stresses and can also comprise various primary metabolites or their biosynthetic intermediates (*Stringlis, De Jonge & Pieterse, 2019*).

Coumarins biosynthesis can be induced in various environments. It involves a complex glycosylation modification that can facilitate adaptations to environmental changes by promoting the stability and biological activity of coumarins (*Garcia et al., 1995*; *Valle et al., 1997*; *Baillieul, de Ruffray & Kauffmann, 2003*; *Gachon, Baltz & Saindrenan, 2004*; *Shimizu et al., 2005*; *Carpinella, Ferrayoli & Palacios, 2005*; *Graña et al., 2017*). For example, plant roots secrete coumarins such as scopoletin, esculetin, and daphnetin under conditions of iron deficiency (*Rajniak et al., 2018*). Moreover, the roots and secretions of iron-deficient *A. thaliana* contain simple coumarins. During the growth and development of *A. thaliana*, the secretion of aesculetin, scopoletin, isoephedrine, and methoxycoumarin can be stimulated using buffered nutrient solutions with pH levels of 5.5 or 7.5 (*Sisó-Terraza et al., 2016*). Salt-alkaline stress also substantially promotes coumarin accumulation (*i.e.,* scopoletin and scopolin) in *L. barbarum* (*Kang, 2022*). Scopoletin accumulation has also been observed in cells exposed to low temperatures (*Döll et al., 2018*). *Wang (2021a)*; *Wang (2021b)*; *Wang (2021c)* simulated low-temperature and drought-stress treatments using a *Melilotus* system and found that the coumarin content gradually increased as the stress became more severe. In addition, scopoletin, scopolin, umbelliferone, and strigolactone play important ecological roles in drought adaptation (*Harbort et al., 2020*). In another study, drought conditions considerably increased coumarin content in the leaf extracts of *Ficus deltoidea* (*Manurung et al., 2019*). *Sun et al. (2022)* reported that the content of coumarin in *Tamarix taklamakanensis* increased considerably after 14 days of drought treatment and that a similar treatment promoted coumarin accumulation in *Cinnamomum cassia* seedlings (*Zhong et al., 2021*). The leguminous plant *Melilotus albus* is abundant in coumarin and is highly tolerant of extreme environments such as drought, cold, and high salt. *Duan et al. (2021)* studied the expression levels of UGT under drought stress and reported that UGT expression was considerably upregulated under drought stress, increasing coumarin biosynthesis and facilitating adaptation to abiotic stress. *Jian*

*et al. (2020)* reported that two CYP71AJ enzymes (*i.e.,* PS and angelicin synthase) were involved in the biosynthesis of furancoumarin in *P. praeruptorum* and that the activity of these enzymes was strongly affected by drought conditions. Moreover, the transcription levels of CYP71AJ49 and CYP71AJ51 were downregulated under drought conditions.

*Genetic factors.* At present, research into the biosynthetic pathways of autotoxic substances remains ongoing. Genetic studies have shown that the production of plant autotoxic substances occurs under genetic control, and their biosynthesis is affected by various regulatory genes, enzymes, and transcription factors. The plant genome contains thousands of genes, of which only 15%–25% are involved in specialized metabolic pathways. The expression of these genes are regulated by different transcription factors, which thereby affect metabolic pathways by influencing metabolic flux (*Broun et al., 2006*). In addition, many transcription factors involved in the regulation of plant defense mechanisms are members of the ERF, bZIP, MYB, bHLH and WRKY TF superfamilies. Among them, MYC2 transcription factors belong to the bHLH family. For example, the transcription factor NaMYC2 is known to play an important role in the regulation of various biosynthesis pathways involved in tobacco (*Nicotiana tabacum*) defense responses (*Woldemariam et al., 2013*). The content of autotoxic substances vary according to the type of adverse conditions faced by plants, and enzymes control the synthesis of autotoxic substances in response to environmental changes. Many studies have shown that plant specialized metabolic pathways make use of enzyme-specific substrates. However, enzymes modified by one or more amino acids can accept new substrates. If the changed enzyme produces a product that is beneficial to the plant, these gene modifications are considered to be conducive to increasing the synthesis of the enzyme, and thereby change the expression of the responsible gene (*Pichersky & Gang, 2000*). It has been found that different *Melilotus* germplasms showed differences in coumarin content; moreover, the expression of genes related to coumarin synthesis and regulation at the RNA level were also different (*Luo, 2017*). Previous studies have used GS-MS to determine the coumarin content of different varieties of alfalfa plants and in rhizosphere soil, and have generally found that the levels of coumarin in different varieties differed significantly (*Li et al., 2009a*; *Li et al., 2009b*; *Rong, Shi & Sun, 2016*).

*Individual developmental factors.* The content of autotoxic substances varies considerably among different stages of plant development. For example, the accumulation of coumarin in different parts of *H. moellendorffii* Hance was the greatest during the seed harvest period (*Sun, 2019*). Moreover, the coumarin content in the roots of *Angelica pubescens* at different developmental stages showed an obvious pattern of S-type growth, and coumarin content reached a maximum level in October (*Ji, Chen & Li, 2022*). Furthermore, *Li (2022)* analyzed differences in coumarin accumulation in *P. praeruptorum* at different developmental stages. *Li (2022)* found that the content of praeruptorin A, praeruptorin B, and praeruptorin E in roots were highest before bolting and lowest after flowering. In addition, the coumarin content of the fruits and tissues of *L. barbarum* has been found to vary with organ maturity. The content of five different types of coumarin were highest in

green fruits and decreased with fruit maturity (*Huang et al., 2013*). Finally, in alfalfa, the coumarin content of different growth stages was ranked as follows: podding stage >early flowering stage >bud stage >branching stage >seedling stage (*Li, 2005*; *Yuan, 2008*).

*Differences among tissues and organs.* There is strong evidence that the biosynthesis and accumulation of autotoxic substances differs among different organs or tissues. Autotoxic substances produced by different plant parts are complex and diverse, and may be synthesized following activation of specific regulatory and transport pathways in particular organs, tissues, and cells (*Belkheir et al., 2016*; *Pichersky & Gang, 2000*). For example, coumarin accumulation was greater in the roots of *H. moellendorffii* than in other plant parts (*Sun, 2019*). Moreover, the scopoletin content of *L. barbarum* leaves has been found to be higher than that in the root bark (*Kang, 2022*). In another study, autotoxic substances were found to be more abundant in the roots of *S. miltiorrhiza* than in the stem or leaf. Moreover, in *S. miltiorrhiza* dibutyl phthalate and methyl oleate were only detected in the root (*Wang, 2021a*; *Wang, 2021b*; *Wang, 2021c*). In alfalfa, the relative coumarin content of different tissues and organs was ranked as follows: leaf >seed >root >flower >stem (*Chon et al., 2002*; *Wang, 2008*; *Wang, Wu & Zhao, 2017*).

### Differences in coumarins accumulation among different plant varieties under environmental stress

Recent reports have examined the effects of genetic variation, chemical stress (*Verma & Shukla, 2015*), nutrient availability (*Dar et al., 2016*), metal ions (*Ma et al., 2018*) and geographical sources (*Li et al., 2013*) on the abundance of autotoxic substances. The increased accumulation of autotoxic substances is usually the result of a combination of stresses that cause dramatic changes in plant growth, physiology, and metabolism (*Debnath, Pandey & Bisen, 2011*). Traditional breeding methods are usually based on phenotypic analysis and rarely consider the influence of environmental factors. The essence of modern biology is that phenotypic variation in plants results from the combination of genetic variation and environmental modification. Moreover, the response of plants to multiple stresses is unique, and cannot be directly inferred from the study of plant reactions to a single stress stimulus. Genotype and environmental factors should therefore be considered together (*Luo, 2017*). Many studies have shown that there are many forms of genetic control that affect autotoxic substance accumulation in *A. thaliana*, and that these factors generate significant differences in morphology, physiological, biochemical responses, and gene expression between different varieties of the same species (*Kang, 2022*). Furthermore, both primitive citrus (*Citrus reticulate*) and wild citrus have high concentrations of total amino acids. This is significant since when plants are exposed to adverse environmental conditions, higher levels of phenylalanine, tyrosine, and tryptophan promote the rapid biosynthesis of phenolic compounds (*Killiny & Hijaz, 2016*). Consistent with this finding, both forms of citrus synthesize coumarins faster than cultivated citrus (*Zaynab et al., 2018*; *Hussain et al., 2019*; *Rao et al., 2021*). Moreover, *Hui (2022)* found that when citrus plants (*Citrus sinensis*) were subjected to iron deficiency stress, genes related to the coumarin synthesis pathway of were differentially expressed. Here, the authors found a large difference in the synthesis and secretion of coumarin in response to iron deficiency. In addition, different

plant genotypes perform differently under drought stress (*Li et al., 2020*). Accordingly, two genotypes of *A. hypogaea* seedlings subjected to drought conditions showed differences in specialized metabolite content. Specifically, the coumarin content in GG7 genotype seedlings was significantly higher than in TG26 seedlings (*Patel, Fatnani & Parida, 2021*) (Table S4).

## CONCLUSIONS

Specialized metabolites that cause plant autotoxicity include phenols, terpenoids, and nitrogenous organic compounds. Phenolic coumarins are the main autotoxic substance affecting alfalfa. Coumarins exert autotoxicity on the next generation by inhibiting seed germination and seedling growth, suppressing photosynthesis, affecting the accumulation of osmotic regulators and plant antioxidant capacity, increasing the degree of membrane lipid peroxidation in plant cells, and affecting hormone levels.

The synthesis and accumulation of coumarins is the result of plant–environment interactions that are affected by environmental and individual factors (*i.e.,* genetic factors, individual developmental factors, and tissue–organ differences). Moreover, numerous differences in coumarin type and content has been observed in different plant parts and at developmental stages in alfalfa. Coumarin biosynthesis involves three distinct stages. First, the formation of the coumarin nucleus, followed by olefination, hydroxylation and cyclization, and subsequently structural modification. Some studies have focused on the mining of information-characterizing enzymes and genes active at the coumarin nuclear formation stage. These studies have identified a series of key enzymes and genes, including PAL, C4H, 4CL, HCT, COMT, COSY, F6'H1, CCoAOMT1, BGA, CYP450, and MDR. Orthohydroxylation is the next key step in coumarin biosynthesis, and PS, COSY, and S8H are reportedly involved in the orthohydroxylation, isomerization, and lactonization of coumarin. Finally, UGTs are responsible for the glycosylation modification of coumarin, and there is evidence that a gene known as MaUGT may be involved in coumarin biosynthesis at this step.

## FUTURE PROSPECTS

In recent years, research on autotoxicity has mainly focused on isolating and identifying plant autotoxic substances and their respective mechanisms of autotoxicity. Conversely, the relationship between the synthesis and accumulation of autotoxic substances and autotoxicity itself needs to be further explored. The production and accumulation of coumarins can be achieved *via* the differential regulation of the expression of biosynthetic genes stimulated by interactions between plant growth and development, tissue differentiation, and external stimuli. Important questions still remain, for instance, under environmental conditions, what are the differences in coumarin anabolism—including carbon and nitrogen metabolism, shikimic acid metabolism, and phenylpropanoid metabolism—among germplasms with different coumarin content? These questions need to be further explored in the future.

At present, a small number of key enzymes, genes, and transcription factors involved in the metabolic pathways related to coumarin synthesis have been identified, and some of these have been functionally verified. However, research on these enzymes and genes mainly focuses on the stage where the coumarin nucleus is formed. Thus, information regarding the enzymes, genes, and transcription factors controlling the gene expression downstream of this stage remains to be excavated.

### Funding
This work was supported by the National Natural Science Foundation of China—Regional Science Foundation Project (NO. 32160330). The funders had no role in study design, data collection and analysis, decision to publish, or preparation of the manuscript.

### Grant Disclosures
The following grant information was disclosed by the authors:
National Natural Science Foundation of China—Regional Science Foundation Project: 32160330.

### Competing Interests
The authors declare there are no competing interests.

### Author Contributions
- Bei Wu conceived and designed the experiments, performed the experiments, analyzed the data, prepared figures and/or tables, authored or reviewed drafts of the article, and approved the final draft.
- Shangli Shi conceived and designed the experiments, performed the experiments, analyzed the data, prepared figures and/or tables, authored or reviewed drafts of the article, and approved the final draft.
- Huihui Zhang conceived and designed the experiments, analyzed the data, prepared figures and/or tables, authored or reviewed drafts of the article, and approved the final draft.
- Baofu Lu performed the experiments, analyzed the data, prepared figures and/or tables, and approved the final draft.
- Pan Nan performed the experiments, analyzed the data, prepared figures and/or tables, and approved the final draft.
- Yun A. performed the experiments, analyzed the data, prepared figures and/or tables, and approved the final draft.

### Data Availability
This is a literature review.

## Supplemental Information

Supplemental information for this article can be found online at http://dx.doi.org/10.7717/peerj.16508#supplemental-information.

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
