# Peer review of "Anabolic metabolism of autotoxic substance coumarins in plants"

_PeerJ, doi:10.7717/peerj.16508_

## Round 0.1 · original submission · Major Revisions

As the editor, I recommend implementing significant revisions in accordance with the reviewers' comments to enhance the overall quality and relevance of the manuscript. These revisions will strengthen the paper and address the concerns raised by the reviewers

**Language Note:** The review process has identified that the English language must be improved. PeerJ can provide language editing services - please contact us at copyediting@peerj.com for pricing (be sure to provide your manuscript number and title). Alternatively, you should make your own arrangements to improve the language quality and provide details in your response letter. – PeerJ Staff

Reviewer 1 ·

Basic reporting

This is an interesting study and the authors have collected a unique dataset using various sources and data bases. Overall, the information presented is useful for understanding the anabolic metabolism of coumarins in alfalfa. However, there are several areas in the MS that deserve improvements especially English writing, as for example I have mention below. Please add recent references if convenient. In conclusion part, please also suggest further studies and experiments that can help in deciphering the exact mechanism of biosynthesis and regulation of coumarins in alfalfa.
Abstract Background Line 1:
Autotoxicity is an intraspecific manifestation of allelopathy in species. (in what species? Planst?)
Abstract Methodology line 5, 6:
developmental stages and tissue parts, and their autotoxicity mechanism (or autotoxicity mechanisms?)
Abstract Methodology line 7:
Synthetic or biosynthesis?
Abstract Result line 6:
key enzymes in this progress (or in this process or biosynthesis pathway/metabolic pathway?)
Abstract Conclusion line 2:
for create alfalfa germplasm? Or to create?
Introduction:
Line 51-53 (please rewrite sentence)
In the 1980s, Rice et al. defined the allelopathy as the phenomenon that contemporary donor plants continuously release some secondary metabolites into the soil, directly or indirectly inhibiting the growth of themselves and surrounding plants. (rewrite sentence)
Line 53-55
Autotoxicity is an intraspecific manifestation of allelopathy in species (which species?), which is widely distributed in Solanaceae, Cucurbitaceae, Umbelliferae, Araliaceae, Poaceae and Leguminosae plants.
Line 85
the metabolic pathway of coumarins synthetic or synthesis?
Line 87-89 (please rewrite sentence)
In order to provide a theoretical basis for further elucidating the autotoxicity mechanism and anabolic mechanism of coumarins, it is of great significance to create alfalfa germplasm with low or no autotoxic substances.
Survey methodology
Line 155
Coumarins are the main autotoxic substances in alfalfa.(Coumarins were found to be the main?)
Line 159
metabolic pathwayand (please give proper space)
Line 194
it has been found that the types and contents of coumarins (in plants?) are affected by cultivars, parts, developmental periods and environmental conditions. (please be specific).
Line 195-96 (please rewrite)
Nie & Zhao (2021) determined the content of coumarins in different parts of six pummel (Citrus grandis) fruits varieties, all the eleven coumarins were detected. (all the eleven coumarins were detected…not clear!).
Line 255-58 (please rewrite especially after seed germination…)
Zheng et al. (2018) also reported that the four substances could inhibit the growth of alfalfa in different degrees by adding exogenous autotoxic substances (cinnamic acid, hydroxybenzoic acid, coumarin and tricin), and the seed germination was inhibited, seedlings fresh weight decreased, and the root activity decreased.
Giver proper spacing (for example Line 269 and 316)
nutrients ( Yang H et al, 2021 ).
Line 348 (is mainly caused?) (please rewrite)
The oxidative damage of coumarin treatment on alfalfa seedlings is mainly (caused?) by increasing the content of H2O2, O2-and MDA, and reducing the activity of SOD and glutathione reductase.
Line 527 (please rewrite)
catalyzes the hydrolysis of β-glucosidase bonds between carbohydrates and coumarin core (hydrolysis of β-glucosidase bonds or hydrolysis of glycosidic bond?)
Line 543-44 (substances produced by environmental factors and effecting) (please rewrite)
secondary metabolites, which are substances produced by interacting with the environment and affecting the expression of biosynthetic genes.
Line 715 (please correct repetitions)
In recent years, the research on autotoxicity mainly has focused mainly on the isolation and

Experimental design

The study is well designed.

Validity of the findings

Findings are valid.

Additional comments

Please include more recent references.

Annotated reviews are not available for download in order to protect the identity of reviewers who chose to remain anonymous.

Reviewer 2 ·

Basic reporting

This is a review article but it is written if it is a research article. English language must be improved. The topic has its importance but article must be written in a proper format.

Experimental design

Check the review articles on this topic and write the article in a proper form.

Validity of the findings

It is review article.

Additional comments

it needs major revisions.

Reviewer 3 ·

Basic reporting

No comment.

Experimental design

No comment.

Validity of the findings

No comment.

Additional comments

The manuscript of Bei Wu and colleagues is a review entitled “Anabolic metabolism of the autotoxic substance coumarins in alfalfa”. Overall, if the information provided is of interest, the first problem the reader encounters is that the part dedicated to alfalfa within the document is rather low and diluted. Second, the document mostly focusses on coumarins. From these observations, the title of the manuscript should be changed. In addition, there are several redundancies between the paragraphs. Last, most of the provided information appears like a list rather than a proper discussion. The document needs extensive modifications in order to render it easier to read. Below are points that need to be addressed.

- A lot of information is given, notably on the type of coumarins found in several different plant species. The main problem here is that it looks like a list of coumarins that lacks organization. For instance, what is the phylogenetic relationship between all the described plants? Are there some correlations between the plant species and the type of coumarins? etc. Beside the information given as a list, it is difficult to get what is the main take home message for the readers.

- There are too many redundancies between the paragraphs 1.3, 2, 2.1 and 2.2.

- Throughout the whole document, it is more appropriate to use “specialized” metabolism/metabolites than “secondary metabolism/metabolites.

- In several instances it is difficult to get to which coumarin the authors are referring to (paragraphs 2.1 and 2.2s).

- Lines 39/41. Give some details on how “plays an important role in improving the ecological environment”.

- Lines 90/102. This paragraph is not very useful for the readers. I suggest removing it. If the authors prefer to keep it, I strongly recommend the authors not to mention Sci-Hub in their review. This is because the use of the search engine “Sci-Hub” is forbidden in several countries and several publishers have launched lawsuit against Sci-Hub.

- Line 158. What the authors mean by “scientifically known”? Do the authors mean the “international nomenclature” that is 2H-1-benzopyrane-2-one?

- Lines 158/160. Unlike what is stated, Coumarin was first isolated in 1820 by Vogel and colleagues from tonka beans (Dipteryx odoranta Wild; Fabaceae family).

- Lines 244/247. It is indicated that “Coumarins are also related to physiological processes”. How? It is most probably indirect for most of the described effects.

- Lines 4141/416. To my knowledge it is unclear how esculetin is synthesized.

- Lines 436/439. It is not true that C4H activity “produce intermediate products such as coumaric acid, caffeic acid, and ferulic acid.”.

- Line 460. What is “BGA”? It is missing Figure 4.

- Lines 516/520. Fraxetin and siddereting glycosides are important coumarins for plant iron nutrition. These compounds and the enzymes necessary for their biosynthesis should be added Figure 4.

- Lines 521/538. There are UGT/BGLU enzymes that were clearly characterized in Arabidopsis thaliana and that are missing in this paragraph.

- Lines 633/663. There are too many redundances.

Minor points:
- Line 40. “wide” instead of “wider”.
- Line 85. “synthesis instead of “synthetic”.
- Line 132. The reference is not properly indicated.
- Lines 279, 343 “0.0342mM”? why not using micromolar units?
- Line 411. C2H is missing Figure 4.
- Line 570. Remove “normally”.

---

## Round 0.2 · accepted · Accept

I have reviewed it myself. The authors have significantly improved the manuscript according to reviewer comments. This manuscript is ready for publication.